# The Role of the Transsulfuration Pathway in Non-Alcoholic Fatty Liver Disease

**DOI:** 10.3390/jcm10051081

**Published:** 2021-03-05

**Authors:** Mikkel Parsberg Werge, Adrian McCann, Elisabeth Douglas Galsgaard, Dorte Holst, Anne Bugge, Nicolai J. Wewer Albrechtsen, Lise Lotte Gluud

**Affiliations:** 1Gastro Unit, Copenhagen University Hospital Hvidovre, 2650 Hvidovre, Denmark; lise.lotte.gluud.01@regionh.dk; 2Bevital AS, 5021 Bergen, Norway; adrian.mccann@bevital.no; 3Global Drug Discovery, Novo Nordisk A/S, Novo Nordisk Park, 2760 Måløv, Denmark; edg@novonordisk.com (E.D.G.); doho@novonordisk.com (D.H.); azbu@novonordisk.com (A.B.); 4Clinical Proteomic Group, NNF Center for Protein Research, Faculty of Health and Medical Sciences, University of Copenhagen, 2200 Copenhagen, Denmark; nicolai.albrechtsen@sund.ku.dk; 5Department of Clinical Biochemistry, Rigshospitalet, University of Copenhagen, 2100 Copenhagen, Denmark; 6Department of Biomedical Sciences, Faculty of Health and Medical Sciences, University of Copenhagen, 2200 Copenhagen, Denmark

**Keywords:** cystathionine β-synthase/cystathionine γ-lyase (CBS/CSE) system, glutathione, H_2_S production, liver fibrosis, non-alcoholic steatohepatitis, sulfur metabolism

## Abstract

The prevalence of non-alcoholic fatty liver disease (NAFLD) is increasing and approximately 25% of the global population may have NAFLD. NAFLD is associated with obesity and metabolic syndrome, but its pathophysiology is complex and only partly understood. The transsulfuration pathway (TSP) is a metabolic pathway regulating homocysteine and cysteine metabolism and is vital in controlling sulfur balance in the organism. Precise control of this pathway is critical for maintenance of optimal cellular function. The TSP is closely linked to other pathways such as the folate and methionine cycles, hydrogen sulfide (H_2_S) and glutathione (GSH) production. Impaired activity of the TSP will cause an increase in homocysteine and a decrease in cysteine levels. Homocysteine will also be increased due to impairment of the folate and methionine cycles. The key enzymes of the TSP, cystathionine β-synthase (CBS) and cystathionine γ-lyase (CSE), are highly expressed in the liver and deficient CBS and CSE expression causes hepatic steatosis, inflammation, and fibrosis in animal models. A causative link between the TSP and NAFLD has not been established. However, dysfunctions in the TSP and related pathways, in terms of enzyme expression and the plasma levels of the metabolites (e.g., homocysteine, cystathionine, and cysteine), have been reported in NAFLD and liver cirrhosis in both animal models and humans. Further investigation of the TSP in relation to NAFLD may reveal mechanisms involved in the development and progression of NAFLD.

## 1. Introduction

Non-alcoholic fatty liver disease (NAFLD) is the most common chronic liver disease and is present in 25% of the population worldwide [1]. The rapid increase in prevalence during recent years parallels the increasing occurrence of obesity and the metabolic syndrome. NAFLD is a disease continuum that ranges from simple steatosis to non-alcoholic steatohepatitis (NASH), which is characterized by inflammation and hepatocyte damage. NASH can lead to fibrosis and eventually to cirrhosis, liver failure, and hepatocellular carcinoma. The primary characteristic of NAFLD is the hepatic accumulation of lipids, mainly triacylglycerols (TAGs), due to the increased influx of free fatty acids (FFA) [2]. Dietary overload is believed to be the main underlying driver, but the molecular mechanisms behind the lipotoxicity remain unclear [3,4,5].

The transsulfuration pathway (TSP) is coupled to the production of the antioxidant glutathione (GSH) and the signal molecule hydrogen sulfide (H_2_S), and both have been linked to the pathogenesis of NAFLD [6,7,8,9]. The large number of patients with NASH-related end-stage liver disease, and the emerging pharmacological treatment options, means that there is an urgent need for valid and reproducible biomarkers of disease development and progression.

Several scores have been developed to diagnose hepatic steatosis [10,11,12]. The scores combine different standard blood tests, such as liver enzymes and bilirubin, with markers of insulin resistance and body weight. The scores have been assessed in different populations using diagnostic imaging or histology as the gold standard and there are no studies showing the validity in larger unselected groups. There are no biomarkers which may be used to diagnose NASH but several which can indicate the degree of fibrosis, including the Fibrosis-4 Index and the NAFLD Fibrosis Score [12,13,14]. These scores include standard blood tests such as alanine amino transferase and aspartate amino transferase and are accurate in the exclusion of fibrosis and the identification of patients with cirrhosis but not in the grading of mild to moderate fibrosis. Other fibrosis biomarkers, including the Enhanced Liver Fibrosis test and procollagen III and IV, also accurately identify patients without fibrosis and patients with cirrhosis [12,15,16]. However, at present, a liver biopsy is required to diagnose NASH. Repeated biopsies in a large group of patients are resource-demanding and the invasive procedure carries a risk of complications. Inexpensive, valid, and reproducible non-invasive biomarkers are necessary for prognostication, monitoring, and evaluating the response in patients undergoing treatment.

Metabolites in the TSP have been shown to predict steatosis and fibrosis in alcoholic liver disease [17], and the TSP has been suggested as a potential drug target in NAFLD and portal hypertension [18,19]. Thus, TSP metabolites may have potential as NAFLD biomarkers, and as our understanding of its relation to NAFLD deepens, support for trials investigating pharmaceutical interventions targeting the TSP may gather strength. This review will focus on the TSP, its regulation and relation to other pathways, and what is known about its implications for NAFLD.

## 2. Description of the Transsulfuration Pathway

The TSP involves the conversion of homocysteine to cysteine via the intermediate cystathionine and plays a key role in sulfur metabolism and the redox environment of cells. The pathway is the only route for biosynthesis of cysteine. The first step is catalyzed by the vitamin B6-dependent enzyme cystathionine β-synthase (CBS), using homocysteine and serine as substrates to form cystathionine in a condensation reaction (Figure 1). The second step is a hydrolyzation reaction catalyzed by the vitamin B6-dependent cystathionine γ-Lyase (CSE), using cystathionine as substrate and producing cysteine and α-ketobuyrate (aKB).

The figure shows the TSP (highlighted in red) and related pathways, including the methionine and folate cycles, glutathione production, cysteine oxidation, and choline oxidation pathway.

For enzyme abbreviations (in red text colour), please see Table 1.

### 2.1. One-Carbon Metabolism

The TSP is part of the one-carbon metabolism (OCM), which is a universal metabolic process that serves to activate and transfer one-carbon (1C) units for the biosynthesis of multiple molecules including purine and thymidine and remethylation of homocysteine [20]. The methionine cycle, folate cycle, and choline oxidation pathway, which are all upstream of the TSP, are also part of OCM. Folate metabolism is central in OCM as many of the different molecules in the folate complex function as carriers of 1C units. The liver is a crucial organ for OCM, with a high capacity for the different processes in OCM [20].

The TSP is closely linked to methionine metabolism, which involves the conversion of methionine to homocysteine. The process is reversible as homocysteine can be converted back to methionine catalyzed by vitamin B12-dependent enzyme methionine synthase (MS) using 5-methyltetrahydrofolate as methyl-donor or by the enzyme betaine-homocysteine S-methyltransferase (BHMT) with betaine as methyl-donor. The latter process is folate-independent and the expression of BHMT is restricted to the kidneys and liver [21].

The remethylation of homocysteine is important because methionine is a substrate for methionine adenosyltransferase (MAT) that synthesizes S-adenosyl-methionine (SAM). SAM, a methyl carrier, is a common enzymatic cofactor and is involved in epigenetics and biosynthesis of phosphatidylcholine, creatinine, and polyamine [22,23,24].

### 2.2. Downstream of the Transsulfuration Pathway

The conversion of homocysteine to cysteine is irreversible. Due to the one-way direction of the TSP, all downstream pathways of the TSP are inherently part of cysteine metabolism. Cysteine, which is a non-essential amino acid, is central to sulfur metabolism and a precursor of important metabolites including GSH, H_2_S (Figure 2), sulfate, and taurine.

In healthy individuals, the steady-state metabolic condition involves a balance between the intake of sulfur from methionine (Figure 1) and the metabolism of sulfur from homocysteine through the TSP. Nearly all sulfur from methionine is transferred to cysteine [25]. Disruption of the TSP has been linked to several diseases including homocystinuria, Huntington’s chorea, and vascular dysfunction, as well as ageing-related changes [26].

### 2.3. Allosteric and Posttranslational Regulation of the Transsulfuration Pathway Flux

The pyridoxal 5-phosphate (PLP—the active form of vitamin B6)-dependent enzymes, CBS and CSE, regulate the metabolic flux through the TSP. Both are highly expressed in the liver and, to a lesser extent, in various other tissues (Table 1). In the liver, CBS and CSE are mainly expressed in the hepatocytes but can also be found in the hepatic vascular system [27,28,29]. CSE is also found in hepatic stellate cells and the terminal branches of the blood vessels in the intrahepatic portal triads [29,30]. The high expression of these enzymes affords the liver a high capacity for transsulfuration.

CBS is tightly regulated by S-adenosylmethionine (SAM), a methyl donor in most transmethylation reactions, through allosteric activation. SAM binds non-covalently to a heme group in CBS, controlling the redox sensitivity of CBS, and stabilizes the enzyme [31]. Decreasing concentration of SAM therefore leads to low CBS activity. The activity of CBS is also stimulated by S-glutathionylation under oxidative stress, thus creating a positive feedback loop [32]. Conversely, SAM inhibits the vitamin B2 (FAD)-dependent methylenetetrahydrofolate reductase (MTHFR) and BHMT, with the net effect being decreased remethylation of homocysteine to methionine and an increase in homocysteine metabolism through the TSP [33]. Thus, the TSP is important in controlling sulfur levels and the degradation of methionine, homocysteine, and cysteine. Other regulators of CBS activity include carbon monoxide (CO) and nitrogen oxide (NO), both by binding to the heme group in CBS, inhibiting its activity [34].

Unlike CBS, the activity of CSE is mainly regulated at the transcriptional level. However, posttranslational modifications such as phosphorylation and sulfhydration can also regulate CSE activity [35,36]. The enzymes in the TSP and related pathways are listed in Table 1.

### 2.4. Importance of Folate and Vitamin B6 and B12 Status in One-Carbon Metabolism and the Transsulfuration Pathway

As cofactor for CBS and CSE, adequate vitamin B6 status is important for the functioning of the TSP. Vitamin B6 deficiency causes reduced activity of CBS and CSE [37], with CSE activity being most affected [38,39,40]. In human studies evaluating moderate B6 insufficiency, no changes in the TSP flux were found [39]. However, increased cystathionine plasma concentration is observed, but with no changes in cysteine or GSH concentrations [41]. The TSP flux is largely maintained due to the increased cystathionine levels [39].

Extensive research has also demonstrated the importance of folate, vitamin 6, and vitamin B12 status in regulating OCM [20,42,43,44,45,46]. The classical symptom of folate and vitamin B12 deficiency is megaloblastic anemia caused by inhibition of DNA synthesis in erythropoietic cells in the bone marrow [46].

### 2.5. Transcriptional Regulation of Enzymes in the Transsulfuration Pathway

The expression of CSE is affected by stimuli such as oxidative stress, endoplasmic reticulum (ER) stress, Golgi stress, inflammation, and nutrient deprivation, whereas CBS is expressed at a less variable rate [26,47]. In the liver, the CSE expression is regulated through the farnesoid X receptor (FXR) and the G-protein-coupled bile acid receptor 1 (GPBAR-1) [35,48]. Upon activation, the FXR binds to an FXR-responsive element in the 5′ flanking region of the CSE promoter, while GPBAR-1 recruits CREB, a cyclic AMP responsive element (CRE) binding protein that binds to sites on the CSE promoter. [35,48].

The transcription factor specificity protein 1 (SP1) is also involved in the regulation of CSE expression. SP1 acts by binding to the CSE promoter, increasing CSE expression. The increased expression enhances H_2_S production by CSE, which causes sulfhydration of the p65 subunit of the transcription factor NF-κβ, which facilitates its recruitment to anti-apoptotic genes [49].

The protein nuclear factor erythroid 2-related factor 2 (NRF2), which regulates the expression of antioxidant proteins, increases the expression of CBS as well as CSE by binding to the promoters [50,51]. NRF2 activators also regulate GSH levels through de novo synthesis and recycling [52,53].

In cardiomyocytes, H_2_S and homocysteine affect CBS and CSE expression through feedback regulation [54]. A similar mechanism may exist in hepatocytes. The transcriptional regulation of the TSP appears to be a comprehensive multileveled network. The regulators (FXR, GPBAR-1, SP1, and NRF2) are also involved in lipid, glucose, and bile acid metabolism, inflammation, and oxidative stress [53,55,56,57], which in turn suggests that the expression of TSP enzymes is regulated in response to the same cellular and metabolic challenges.

### 2.6. Hydrogen Sulfide Production Is Inherently Coupled to the Transsulfuration Pathway

H_2_S is a colorless gas that has emerged as an important signaling molecule alongside NO and CO. It is an important mediator of cell functions and is involved in physiological processes such as inflammation, apoptosis, vasorelaxation, and neuromodulation [58]. It is also thought to increase the production of GSH [59]. One possible mechanism of H_2_S action is by posttranslational modification of the cysteine residues on target proteins, yielding a hydropersulfide moiety (-SSH) or polysulfide in a process known as S-sulfhydration or S-persulfidation [60]. An animal study found that a substantial number of proteins in the liver are S-sulfhydrated and that S-sulfhydration alters protein function [36]. In humans, H_2_S is mainly produced via CBS and CSE (Figure 2), but also by 3-mercaptopyruvate sulfurtransferase (MPST) as part of the cysteine catabolic pathway [61].

The extent to which CBS and CSE are active in the production of cysteine or H_2_S depends on several factors. Substrates binding to the heme group of CBS can direct the activity towards H_2_S production [62]. High CO levels inhibit CBS activity, causing an increase in homocysteine and subsequently H_2_S production [63]. Availability of substrates is also important since CBS has a higher affinity for serine, and CSE a higher affinity for cystathionine compared to cysteine. If serine levels are elevated, the production of cystathionine will increase, and if cystathionine is elevated, the production of cysteine will increase. A study using a NAFLD model and liver cells surprisingly found that the H_2_S-producing enzyme MPTS suppressed H_2_S production primarily through downregulation of CSE [64]. Partial knockout of MPST, either via adenovirus-mediated short hairpin RNA (shRNA) delivery or heterozygous deletion, significantly upregulated hepatic CSE expression and increased hepatic H_2_S levels in high-fat-diet (HFD)-fed mice and in FFA-treated L02 cells. This emphasizes the complexity of the metabolic processes involved in TSP regulation and H_2_S levels.

### 2.7. Pathways Involving the Conversion of Cysteine to Glutathione, Taurine, and Sulfate

Cysteine has several potential fates beyond protein synthesis and H_2_S production. The TSP is linked to the production of the major antioxidant GSH (Figure 1). Accordingly, TSP metabolic flux is important for the redox environment. GSH imbalance is linked to several diseases, including type 2 diabetes, cancer, and pulmonary fibrosis [65]. Nearly half of the intracellular GSH pool in human liver cells is derived from homocysteine via the TSP [66]. Oxidative stress increases the flux through the TSP, which subsequently leads to increased GSH production [67].

In addition to GSH, cysteine is converted to the sulfur-containing molecule taurine. The first reaction involves oxidation of the thiol in the cysteine molecule by cysteine dioxygenase (CDO) to form cysteinesulfinate, which is further degraded to taurine, sulfate, and pyruvate. Taurine is important for several biological functions, including muscle functioning, calcium homeostasis, and neuro- and immunomodulation [68]. Taurine also reduces the secretion of apolipoprotein B100 and lipids from liver cells, probably by inhibition of TAG and cholesterol ester synthesis [69]. CDO uses cysteine as substrate and can therefore modulate H_2_S and GSH production, and CDO levels are increased in response to high-intracellular cysteine through a block of ubiquitination (the bonding of ubiquitin to a substrate protein, marking the protein for degradation) [70]. CDO is highly expressed in the liver. Mice lacking CDO have elevated cysteine, GSH, and H_2_S levels, but the liver phenotype has not been described [71,72,73]. The expression of CDO in murine liver increases by up to 45-fold and the activity by up to 10-fold depending on cysteine availability [25].

## 3. Alterations in the Transsulfuration Pathway Linked to NAFLD in Experimental and Animal Models

NAFLD is a complex disease, with multiple molecular events and altered metabolic pathways being part of the pathogenesis. The TSP plays a role in many of these events and has been linked to steatosis, insulin resistance, oxidative stress, ER stress, inflammation, and portal hypertension [6,19,66,74,75,76,77,78]. The link between the TSP and oxidative stress is believed to be mediated through its regulation of GSH production. In CBS-deficient mice, plasma homocysteine levels are increased and steatosis, oxidative stress, and fibrosis develop in the liver [74,75,79]. The lack of CBS appears to upregulate the expression of genes involved in ER stress, hepatic lipid homeostasis, and genes associated with hepatic steatosis [79,80], while knockout of CSE causes reduced hepatic lipolysis [76].

In high-fat-diet models of NAFLD, the hepatic expression of CBS and CSE has provided conflicting results [81,82,83,84]. This could reflect the differing dietary compositions but also the timing of the assessment. It is likely that there is an early adaptive upregulation in the expression of CBS and CSE in response to the increased influx of FFA, and subsequently, the TSP is downregulated or dysfunctioning. Speculatively, enzyme expression could be upregulated with increased hepatic insulin signaling in the initial phase, followed by downregulation with the onset of insulin resistance; however, there are currently no data to support this. The increase in homocysteine caused by the disruption of the TSP also appears to have intrinsic pathophysiological effects. A study evaluating homocysteine-induced ER stress in human hepatocytes found activation of the unfolded protein response (UPR) and the sterol regulatory element-binding proteins sterol regulatory element-binding proteins (SREBPs) [85]. Mice with diet induced-homocysteinemia developed hepatic steatosis, suggesting that homocysteine plays a role in NAFLD development.

### 3.1. Reduced Production of H_2_S May Be Involved in NAFLD

The liver is a major organ for endogenous H_2_S production and clearance. H_2_S is involved in mitochondrial functioning, insulin sensitivity, lipoprotein synthesis, and glucose metabolism in the liver [6]. H_2_S also protects against ischemic liver injury [30,86,87]. An animal model of NAFLD showed that H_2_S production was reduced in a high-fat and methionine- and choline-deficient diet [88]. The reduced production could be explained be decreased expression of CSE and CBS, but also by the unavailability of methionine and choline [88]. It has been speculated that exogenous H_2_S prevents NASH development in mice by decreasing inflammation and oxidative stress [89]. The reason that H_2_S synthesis is impaired in NAFLD remains unclear but is probably due alterations in the TSP, since CBS and CSE are responsible for most H_2_S production.

H_2_S modulates hepatic steatosis through downregulation of the expression of SREBP-1c, the major transcriptional regulator of the enzymes involved in de novo lipogenesis [5], and the downstream lipogenic enzymes, fatty acid synthase (FAS), and acetyl-CoA carboxylase (ACC). The H_2_S mediated protection against oxidative stress and hepatocyte injury may be related to the suppression of the C-Jun N-terminal kinase (JNK) signaling pathway [64,90]. The FXR is involved in lipogenesis and is an activator of CSE expression and H_2_S production. The hepatic expression of SREBP-1c, FAS, and liver X receptor (LXR) was increased in patients with NAFLD, while the expression of FXR was decreased [91]. A phase 3 randomized trial investigating the FXR agonist obeticholic acid for the treatment of non-alcoholic steatohepatitis showed promising results [92]. FXR activation suppresses lipogenesis and decreases SREBP-1c expression [91]. SREBP-1c and its upstream regulators (H_2_S, CSE, and MPST) may therefore represent candidate targets in the treatment of NAFLD.

Dysregulation of TSP and H_2_S production may also play a role in the development of portal hypertension in NASH cirrhosis. Downregulation of the TSP leads to decreased levels of H_2_S, which is a potent vasodilator, and an impaired TSP contributes to increased intrahepatic vascular resistance in rodent models of liver cirrhosis [19]. In agreement with these findings, H_2_S administration leads to relaxation of the portal vein [93].

### 3.2. Taurine May Have Beneficial Effects in NAFLD Models

Taurine, the major end-product of cysteine oxidation, affects lipid metabolism and ameliorates the accumulation of lipids in the liver [94,95]. The capacity for de novo taurine biosynthesis is limited and hepatic taurine deficiency is primarily caused by an insufficient nutritional uptake rather than dysfunction of the endogenous biosynthesis [94]. Most studies performed on taurine and NAFLD have investigated the effect of taurine administration on cell cultures or animals. The anti-oxidative effect of taurine may reduce mitochondrial dysfunction and ER stress [95,96,97]. Furthermore, taurine administration leads to improved GSH production [95]. Knockout of the taurine transporter in mice is associated with hepatocyte apoptosis, inflammation, and liver fibrosis [98]. Administration of taurine in animal models of NAFLD results in decreased steatosis, inflammation, and oxidative stress [95,97].

### 3.3. Long-Term Lipotoxicity Leads to Glutathione Depletion and May Be Involved in NAFLD Development

In silico and in vitro analyses indicate that the initial cellular response to increased hepatic FFAs and hepatic steatosis is an increase in cellular GSH concentration [99,100]. Studies evaluating diet-induced NAFLD animal models have found increased as well as decreased GSH levels in plasma and the liver [7,77,101,102]. The ratio between GSH and glutathione disulfide (GSSG) (GSH/GSSG—an early marker of oxidative stress) was decreased in NAFLD models [7,102]. In addition, an increased influx of FFA for a prolonged period causes depletion of GSH in the livers of rats [77]. Thus, it appears that the biosynthesis of GSH and conversion of GSSG back to GSH is unable to keep pace with the demand to maintain redox homeostasis in the liver, whereby oxidative stress ensues, eventually leading to the development of NASH [7].

## 4. The Transsulfuration Pathway in Human NALFD

Studies investigating alterations in the TSP in humans with NAFLD are limited, but changes in the sulfur metabolism, especially methionine metabolism, in alcoholic liver disease and cirrhosis are well documented [103,104,105,106,107]. Recently, there has been a focus on the TSP. A study evaluating the relation between metabolites in the TSP and histopathology in alcoholic liver disease (ALD) found that cystathionine levels were positively associated with steatosis and fibrosis [17]. In fact, the levels of several of the metabolites in the TSP and OCM pathways differed between individuals with ALD, active drinkers without liver disease, and healthy controls. Most of the metabolites, including cysteine, were increased in ALD, but the downstream metabolite α-aminobutyrate (aABA) and the aABA/cystathionine ratio were decreased, suggesting an impairment of the TSP [17]. These findings concur with an earlier study measuring TSP metabolites in cirrhotic patients with mixed etiology compared to healthy controls [108]. Cysteine is also increased in NAFLD; the highest levels are seen in patients with NASH and/or fibrosis [8,109,110].

### 4.1. Increased Blood Levels of Homocysteine in Human NAFLD

Homocysteine is used in clinical practice in the assessment of folate (vitamin B9) and cobalamin (vitamin B12) deficiency. Some studies have found that circulating homocysteine is increased in NAFLD compared to controls [8,111,112,113,114,115,116,117,118], whereas other studies observe the opposite [119,120,121]. A meta-analysis combining the results of several studies concluded that homocysteine is increased in NAFLD [122]. However, the homocysteine levels are not higher in patients with simple steatosis compared to NASH [111,114,116]. When comparing homocysteine levels and hepatic histopathological findings, no consistent trends have been observed (Table 2).

The variation in the results could reflect multiple factors. For instance, systemic homocysteine levels are regulated by hepatic efflux, renal clearance rates, and increase with age. Other possible confounders include nutritional factors (such as the intake of proteins, minerals, and B-vitamins) and body composition [124,125,126]. Given these considerations, and the association between elevated homocysteine and various inflammatory conditions and markers [127], homocysteine does not appear to be an adequate biomarker for NAFLD.

### 4.2. Genetic Association of the Transsulfuration Pathway with NAFLD

A study including 268 patients who underwent a liver biopsy during surgery found that single-nucleotide polymorphisms (SNPs) in the Glycine N-methyltransferase (GNMT) gene were associated with hepatic levels of the GNMT protein, while this was not the case for SNPs in the MAT1A gene and the MAT protein [128]. The T-allele of 677C>T polymorphism in the MTFHR gene (rs1801133) is associated with elevated tHcy levels [129]. A large cohort study from Sweden found that the T-allele was associated with elevated tHcy levels and cardiovascular multimorbidity [130]. A meta-analysis of the C677T-SNP and the variant A1298C (rs1801131) found that the homozygous TT genotype was associated with an increased susceptibility to NAFLD [131]. An SNP (variant c.1364G>T) in the CTH gene (CSE) has been associated with elevated homocysteine levels, but polymorphisms in the genes for CBS and CSE have not been linked to NAFLD in humans [132].

Only a few studies have reported on the genetic expression of the TSP and related pathways in human liver disease. In contrast, several NAFLD animal studies have reported reduced expression of the enzymes in the TSP and related pathways [39,60,114,115,116,117]. The expression of CBS and CSE is reduced [60,114], while CDO is found to be increased [115].

Reduced mRNA abundance was found for GNMT, MS, BHMT, MAT1A, and CBS in human cirrhotic livers compared to controls [133,134]. The hepatic expression of MPST correlated to the grade of steatosis in patients with NAFLD and was increased compared to healthy controls [64].

In liver biopsies of patients with NAFLD, advanced disease was associated with general DNA hypomethylation, suggesting less transcriptional control [135]. Some genes in the OCM, namely AHCY, MAT1A, and methylenetetrahydrofolate dehydrogenase (MTHFD2), involved in generating methyl groups for methylation, were hypermethylated, which correlated with low expression of the respective genes [135]. Theoretically, the decreased expression of the enzymes involved in homocysteine remethylation to methionine (MS, BHMT) may cause low hepatic concentration of the methyl donor SAM and subsequently hypomethylation of genes in the liver.

### 4.3. Reduced H_2_S Levels May Be Associated with Liver Disease

To date, there are limited studies investigating the association between H_2_S and liver disease. One study found decreased levels of H_2_S in cirrhotic patients of mixed etiologies compared to healthy controls [136], in line with the reduced expression of CBS observed in cirrhosis [133,134]. Likewise, the reduced H_2_S levels correlate with increased portal vein diameter and Child–Pugh Score [136]. Obesity and diabetes are also associated with reduced H_2_S levels [9,137,138], probably related to poor glycemic control [9].

### 4.4. Transsulfuration Pathway Metabolite Concentrations and Cofactor Status in Human NAFLD

The circulating and tissue concentrations of several metabolites related to TSP and OCM may be changed due to altered TSP flux. These metabolites include the amino acids serine, glutamate, and glycine, in addition to α-ketobutyrate, pyruvate, and sulfur-containing taurine, gamma-glutamylcysteine, and GSH (Figure 1). In analyses of amino acids, plasma glutamate levels were found to be increased while circulating concentrations of glycine and serine were decreased in patients with NAFLD [8,139]. Plasma glutamate and glycine were independent predictors of fibrosis, irrespective of insulin resistance [139]. In the setting of the dysregulated metabolism observed in hepatic steatosis, glycine appears to be the limiting factor in de novo GSH production [99]. Plasma GSH is low and the rate of hepatic glutathione turnover is high in patients with NAFLD compared to healthy controls [8,117]. The hepatic expression of several enzymes involved in the synthesis of GSH was reduced in individuals with obesity compared to healthy subjects [99]. Thus, the high GSH turnover is not accompanied by a corresponding increase in GSH production. Another study found a decrease in human liver GSH in steatotic compared to non-steatotic subjects [140]. The GSH/GSSG ratio in the steatotic group was also decreased, indicating impaired regeneration of GSH and/or increased oxidative stress [140]. The decreased levels of circulating GSH in NAFLD correspond to the findings in patients with type 2 diabetes, suggesting a potential link between GSH and metabolic disease [141]. A small, open-label, single-arm pilot study suggested that oral administration of GSH may improve liver enzymes in some individuals with NAFLD [142]. However, caution is needed when interpreting results from studies evaluating GSH and GSSH levels, because GSH and related thiols are notoriously unstable and sensitive to oxidation and or degradation during sample handling and analysis [143].

Circulating taurine levels are elevated in NAFLD compared to healthy controls [144]. However, when comparing NAFLD patients with and without advanced fibrosis, taurine levels were decreased in the former patient group [13]. Thus, systemic taurine may be a predictor of steatosis and inflammation rather than fibrosis. Hepatic taurine levels have also been found to be increased in human NASH liver tissue relative to samples from patients with simple steatosis or normal liver tissue, suggesting a hepatic origin for the changes seen in circulating taurine levels [145,146]. Elevated taurine levels could be due to increased biosynthesis in NASH patients, consistent with increased expression of CDO in an NAFLD animal model [147].

The role of vitamin B6 in human NAFLD has not been well examined. A study found that high vitamin B6 intake was associated with the occurrence of NAFLD as evaluated by controlled attenuation parameter (CAP) [148]. However, the circulating levels of vitamin B6 were not measured. In contrast, some animal models have shown that B6 deficiency causes hepatic lipid accumulation and is prevented by supplementation [149,150,151]. One study has reported reduced levels of circulating vitamin B6 in patients with NAFLD compared to controls [149]. Describing the mechanisms linking alteration in vitamin B6 status to NALFD is challenging as vitamin B6 is a cofactor for around 150 enzymatic activities, but vitamin B6 status has been coupled to type 2 DM and lipid metabolism [152].

The levels of circulating folate and vitamin B12 in human NAFLD have been studied more extensively. A meta-analysis found a tendency towards decreased levels of folate and increased levels of vitamin B12 in NAFLD compared to controls, but the results were not statistically different [122].

## 5. Summary

The TSP is involved in the synthesis and processing of several metabolites that are important for cellular function. These metabolites include homocysteine, cysteine, H_2_S, GSH, and taurine, among many others. Importantly, the TSP plays a central role in redox homeostasis and the modulation of cellular oxidative stress. The liver is a major organ in TSP and related OCM processes.

The TSP enzymes (CBS and CSE) along with the other enzymes involved in OCM are all highly expressed in the liver. Thus, changes in these processes may be involved in liver disease and metabolic alterations. A link between TSP and NAFLD may exist; see Figure 3.

Known factors and comorbidities in the development of NAFLD, such as insulin resistance, type 2 DM, obesity, changed glucose and lipid metabolism, and inflammation, can be coupled to changes in the functioning of the TSP and related OCM processes. Knockout of CBS and CSE in animal models leads to liver steatosis, further strengthening the potential link between the TSP and NAFLD. Results from animal models and human liver disease suggest decreased expression of CBS and CSE. The decreased expression and potential changes in activity of the TSP and OCM enzymes will expectedly cause changes in several metabolites. Studies have found changed levels of methionine, SAM, SAH, homocysteine, cystathionine, cysteine, H_2_S, and GSH in human liver disease.

One of the most important challenges in NAFLD is to find new biomarkers of disease severity and development. The available evidence is not yet clear and additional studies are still needed to evaluate whether TSP or OCM metabolites may be used as biomarkers in NAFLD. Cystathionine could be a potential candidate as it has been shown to predict steatosis and fibrosis in ALD, and it is central in the TSP flux and will increase by reduced activity and low expression of CSE.

Because TSP and OCM are central in many metabolic processes, such as methylation, GSH production, amino acid metabolism, and H_2_S production, the enzymes and metabolites may represent potential candidates for therapeutic targets. Currently, it is difficult to nominate the most evident candidates. Modulation of CBS and CSE activity can potentially affect many processes by changing GSH and H_2_S production.

## 6. Conclusions

With the high expression of the CBS and CSE enzymes in the liver, it is not unexpected to find perturbation of the TSP in human liver diseases. To investigate the potential relationship of the TSP with NAFLD, experiments involving gene knockout and/or gain of function along with histological evaluation could provide more evidence. Despite increasing insights, it remains a challenge to fully comprehend the complex molecular mechanisms of the TSP and related OCM pathways and its relation to NAFLD pathophysiology. Additional knowledge may point to potential drug targets or biomarkers of disease severity and progression. However, it is important to remember that the TSP is only a small part of a large network of interconnected molecular pathways. Pharmacological modulation of the activity of the TSP enzymes could potentially lead to adverse effects that are difficult to predict due the complexity of the metabolic network involved.

TSP and OCM metabolites may emerge as predictors of disease development and progression in NAFLD. With the development of more complex multifactorial models using applying machine learning and deep learning algorithms, TSP and OCM metabolites can potentially be included in such models. Severity scores will likely require several factors/components, including multiple circulating biomarkers, as well as information concerning age, renal function, body composition, and nutritional status (e.g., measures of B-vitamin status).

Finally, we need additional studies in both cell lines, animals and humans, to further investigate the complex TSP and related pathways to unveil their potential as pharmacological targets and biomarkers.

## Figures and Tables

**Figure 1 jcm-10-01081-f001:**
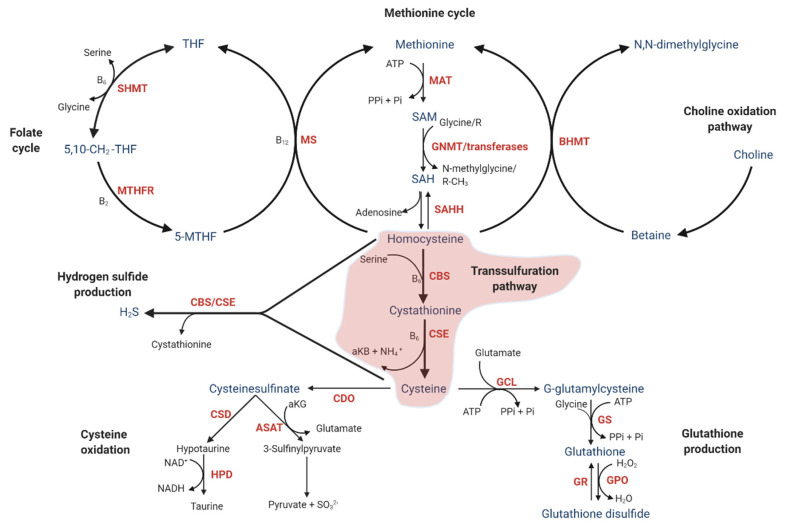
The transsulfuration pathway (TSP) and related pathways. Metabolites: 5-MTHF, 5-methyltetrahydrofolate; 5,10-CH_2_-THF. 5-10-methylenehydrofolate; aKG, α-ketoglutarate; aKB, α-ketobutyrate; ATP; adenosine triphosphate; B_2_, riboflavin; B_6_, pyridoxal 5′-phosphate; B_12_, cobalamin; H_2_S, hydrogen sulfide; NH_4_^+^, ammonia; Pi, inorganic phosphate; PPi, pyrophosphate; SAH, S-adenosylhomocysteine; SAM, S-adenosylmethionine; SO_3_^2−^, sulfite; THF, tetrahydrofolate.

**Figure 2 jcm-10-01081-f002:**
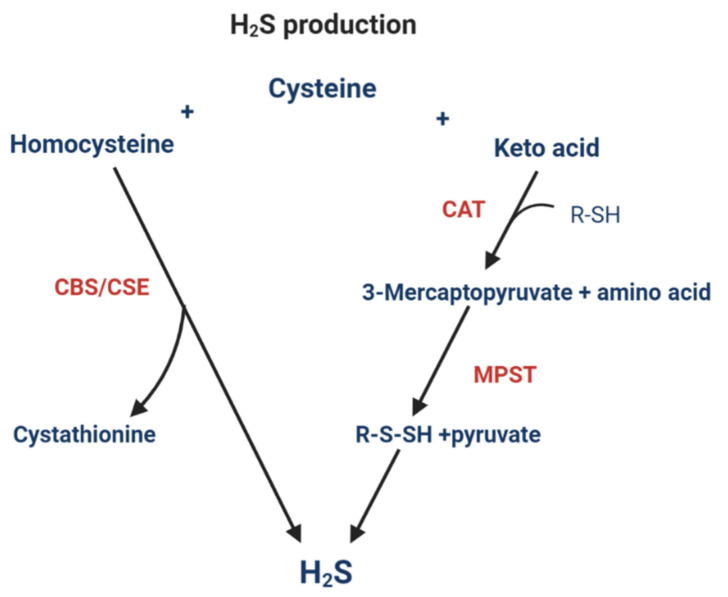
Hydrogen sulfide production pathways. The biosynthesis of hydrogen sulfide (H_2_S) involves the three enzymes, cystathionine β-synthetase (CBS), cystathionine γ-lyase (CSE), and 3-mercaptopyruvate sulfurtransferase (MPST). The process involving H_2_S production via CBS and CSE uses homocysteine and cysteine as substrates. The production of H_2_S via cysteine transaminase (CAT) and MPST uses cysteine and a keto-acid as substrates to form 3-mercaptopyruvate in the first step via CAT, followed by the synthesis of H_2_S from 3-mercaptopyruvate via MPST.

**Figure 3 jcm-10-01081-f003:**
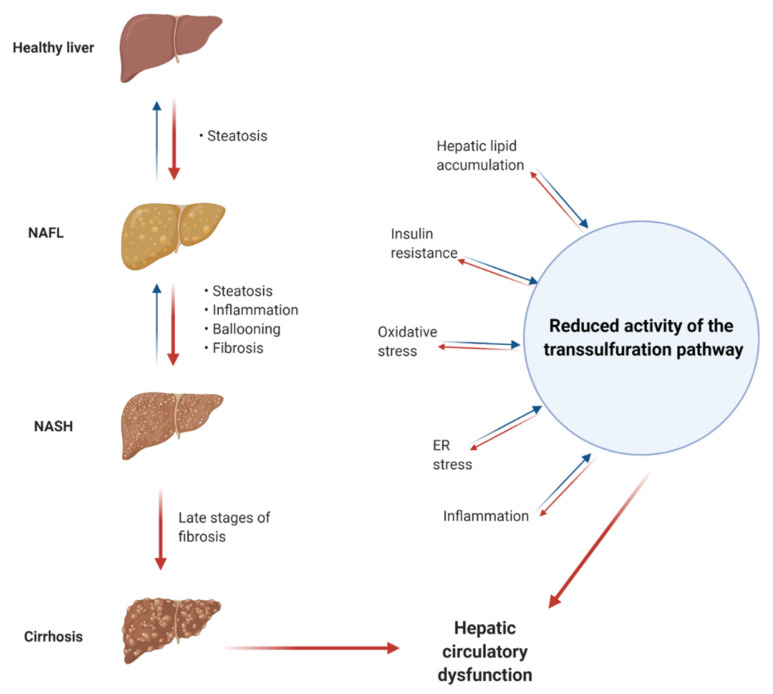
Potential links between the pathogenesis of non-alcoholic fatty liver disease (NAFLD) and reduced activity of the transsulfuration pathway (TSP) and related pathways. NAFL, non-alcoholic fatty liver; NASH, non-alcoholic steatohepatitis.

**Table 1 jcm-10-01081-t001:** An overview of the enzymes in the transsulfuration and related pathways.

Enzyme Name	Gene Name	Cofactor	Structure	Tissue Expression	Cellular Location
Adenosylhomocysteinase (SAHH)	*AHCY*	NAD^+^	Homotetramer 432 amino acids 47.7 kDa	Low tissue specificity. Highly expressed in liver, pancreas and kidney, endocrine tissue, female and male tissue.	Cytoplasma
Betaine-homocysteine S methyltransferase (BHMT)—Two genes	*BHMT* *BHMT2*	Zink	Homotetramer 406/363 amino acids 45.0/40.4 kDa	High tissue specificity. Highly expressed in liver, kidney, and urinary tract.	Cytoplasma
Cystathionine beta-synthase (CBS)	*CBS*	Pyridoxal 5′-phosphat (B6)	Homotetramer 551 amino acids 61 kDA	High tissue specificity. Highly expressed in liver and pancreas. Some expression in heart and brain.	Nucleus/cytoplasma
Cystathionine gamma-lyase (CSE)	*CTH*	Pyridoxal 5′-phosphat (B6)	Homotetramer 405 amino acids 44.5 kDa	High tissue specificity. Highly expressed in liver, female tissue and endocrine tissue. Some in the pancreas, brain, and kidneys.	Cytoplasma
Cysteine dioxygenase (CDO)	*CDO1*	Iron	Monomer 200 amino acids 30.0 kDA	High tissue specificity. Highly expressed in liver and placenta. Some expression in heart, sdipose tissue, brain, and pancreas.	Cytoplasma
Cysteine sulfinic acid decarboxylase (CSD)	*CSAD*	Pyridoxal 5′-phosphat (B6)	Homodimer 493 amino acids 55.0 kDA	Low tissue specificity. Expressed in liver, gastrointestinal (GI) tract, brain female and male tissue, muscle tissue, and adipose tissue.	Cytoplasma
Glutathione peroxidase (GPO)—several genes	*GPX 1–8*		GPX2—Homotetramer 190 amino acids 22.0 kDa	GPO has low tissue specificity. GPX 2 is highly expressed in the liver, gallbladder and GI tract.	Cytoplasma/mitochondrion
Glutathione reductase (GR)	*GSR*	FAD	Homodimer 552 amino acids 56.3 kDa	Low tissue specificity. Highly expressed in liver, pancreas, GI tract, endocrine tissue, kidney, female and male tissue.	Cytoplasma/mitochondrion
Glutathione synthetase (GS)	*GSS*	Magnesium	Homodimer 474 amino acids 52.4 kDa	Low tissue specificity. Highly expressed in brain, endocrine tissue, GI tract, kidney and liver.	Cytoplasma
Glutamate-cysteine ligase (GCL)	*GCLM* *GCLC*		Heterodimer 274 + 252 amino acids 33.7 + 28.1 kDa	Methionine synthase Highly expressed in the liver.	Cytoplasma
Glycine N-methyltransferase (GNMT)	*GNMT*		Homotetramer 295 amino acids 32.7 kDa	High tissue specificity. Highly expressed in liver and pancreas. Some expression in brain, GI tract, and kidney.	Cytoplasma
Methionine adenosyltransferase (MAT)	*MAT1A*	Potasium Magnesium	Homodi- and tertramer 395 amino acids 43.6 kDA	High tissue specificity. Highly expressed in liver, pancreas. Some expression lungs and female and male tissue.	Cytoplasma
Methionine synthase (MS)	*MTR*	Cobalamin (B12) Zink	Monomer- and Dimer 1265 amino acids 140.5 kDa	Low tissue specificity. Highly expressed in pancreas, heart, brain, skeletal muscle and placenta. Expressed at lower levels in lung, liver, and kidney.	Cytoplasma
Methylenetetrahydrofolate reductase (MTHFR)	*MTHFR*	FAD	Homodimer 656 amino acids 74.6 kDA	Low tissue specificity. Highly expressed in female and male tissue, GI tract and kidney. Some expression in liver.	Cytoplasma
Serine hydroxymethyltransferase (SHMT)—two genes	*SHMT1 SHMT2*	Pyridoxal 5′-phosphat (B6)	Homotetramer 486/504 amino acids 53.1/60.0 kDa	High tissue specificity. Highly expressed in liver and kidney. Some expression in lungs, brain, pancreas, and GI tract.	Cytoplasma/mitochondrion

Data were retrieved from: www.proteinatlas.org; https://bgee.org/; https://www.ebi.ac.uk (accessed on 16 November 2020).

**Table 2 jcm-10-01081-t002:** Correlation between histological scores for non-alcoholic fatty liver disease (NAFLD) and homocysteine concentrations.

	Histology				
Study	Steatosis	Fibrosis	Inflammation	Ballooning	NAS
Brochado 2013 [111]	No	No	No	NA	NA
Gulsen 2005 [114]	NA	Positive	NA	NA	Positive
Hirsch 2005 [119]	NA	No	No	NA	No
Lai 2020 [116]	Positive	No	No	NA	Positive
Pylozos 2012 [120]	No	Negative	Negative (portal)	No	NA
Xu 2020 [123]	No	Negative	No	Negative	NA

NA: not assessed. NAS: NAFLD activity score.

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
