# Peer review of "The Role of the Transsulfuration Pathway in Non-Alcoholic Fatty Liver Disease"

_jcm, 2021, doi:10.3390/jcm10051081_

Round 1
Reviewer 1 Report
Overall the review was comprehensively thought out and executed with superior organizational thought.
Quite a lot of information was presented but with a flow that readers can understand.
The figure diagrams were very clean and helpful. The table was quite useful to tie in the entire review.
Some very minor editing throughout the script.
Great review!
Author Response
Thank you for the positive comments. We have revised our paper to improve readability and ensure that the text is free of grammatical errors.
Reviewer 2 Report
Comment to Authors
The paper by Mikkel Parsberg Werge entitled ‘Reduced activity in the transsulfuration pathway may be associated with non-alcoholic fatty liver disease’ is an article that reviews the published literature related to pathway of non-alcoholic fatty liver disease.
Author Response
Thank you for reviewing our paper.
Reviewer 3 Report
In this review, the authors discuss changes in the transsulfuration pathway in regards to NAFLD pathogenesis and focus on research studies that may prove TSP metabolites as potential biomarkers. Because there is a scarcity of reviews that aim to compile studies that demonstrate the relationship between the transsulfuration pathway and NAFLD, this review is quite informative and thus can be of great interest to the scientific community. However, the review does not provide a substantial amount of convincing evidence that supports TSP metabolites as reliable biomarkers. Consequently, the review requires major revisions prior to publishing.
The authors should reconsider the title of the review; perhaps a more general title may be appropriate such as “The Transsulfuration Pathway and Non-alcoholic fatty liver disease”. Because the mentioned findings of the studies summarized within the review are not conclusive and at times offer contradicting findings, the current title may be too focused and somewhat misleading. The authors discussed that there are limited non-invasive approaches for diagnosing NASH and therefore the need for such biomarkers is crucial. The authors may consider discussing in greater detail the available invasive and non-invasive approaches for diagnosing steatosis, steatohepatitis, and fibrosis, including their limitations. This may further highlight the greater need for studying the TSP in NAFLD and considering TSP metabolites as biomarkers. This review would also be greatly improved if the authors discuss more convincing studies that demonstrate TSP metabolites as potential NAFLD biomarkers. The authors may also consider shortening their review on the TSP cycle to do so. The novelty of this review relies on its assessment of the transsulfuration pathway in regards to NALFD. Consequently, a more in-depth assessment and summation of some mentioned studies may also greatly improve the quality of the review (e.g. Lines 265-274).
Author Response
In this review, the authors discuss changes in the transsulfuration pathway in regards to NAFLD pathogenesis and focus on research studies that may prove TSP metabolites as potential biomarkers. Because there is a scarcity of reviews that aim to compile studies that demonstrate the relationship between the transsulfuration pathway and NAFLD, this review is quite informative and thus can be of great interest to the scientific community. However, the review does not provide a substantial amount of convincing evidence that supports TSP metabolites as reliable biomarkers. Consequently, the review requires major revisions prior to publishing.
Response: Thank you for reviewing our paper. We agree that the role of TSP in NAFLD is complex there is no conclusive evidence to support that use of TSP metabolites as biomarkers. We have therefore adjusted our wording to highlight the fact that additional evidence is needed before any definite conclusions can be made.
The authors should reconsider the title of the review; perhaps a more general title may be appropriate such as “The Transsulfuration Pathway and Non-alcoholic fatty liver disease”. Because the mentioned findings of the studies summarized within the review are not conclusive and at times offer contradicting findings, the current title may be too focused and somewhat misleading.
Response: We agree that the title could be viewed as misleading and have therefore changed the title to ‘’The role of the transsulfuration pathway in non-alcoholic fatty liver disease’’.
The authors discussed that there are limited non-invasive approaches for diagnosing NASH and therefore the need for such biomarkers is crucial. The authors may consider discussing in greater detail the available invasive and non-invasive approaches for diagnosing steatosis, steatohepatitis, and fibrosis, including their limitations. This may further highlight the greater need for studying the TSP in NAFLD and considering TSP metabolites as biomarkers.
Response: We have included a paragraph discussing the evidence for biomarkers identifying steatosis, steatohepatitis and fibrosis (page 2, line 59-69).
This review would also be greatly improved if the authors discuss more convincing studies that demonstrate TSP metabolites as potential NAFLD biomarkers. The authors may also consider shortening their review on the TSP cycle to do so.
Response: Unfortunately, we are not aware that there are more convincing studies demonstrating that TSP metabolites are NAFLD biomarkers. This is now mentioned (page 16, line 462-463).
The novelty of this review relies on its assessment of the transsulfuration pathway in regards to NALFD. Consequently, a more in-depth assessment and summation of some mentioned studies may also greatly improve the quality of the review (e.g. Lines 265-274).
Response: We have now described the studies in more detail.
Reviewer 4 Report
In this review, the authors reported the association between transsulfuration pathway and non-alcoholic fatty liver disease. This is very interesting as it could help in identifying and treating patients with NASH that are surging.
The authors should be complimented for this review that is very accurate, well written and describes in east word a complicated molecular pathway.
I have no suggestions for the authors to improve the quality of this review that is already an high level review.
Author Response
Thank you for reviewing our paper.
Round 2
Reviewer 3 Report
Thank you for addressing the suggestions made. I have no further major comments but I do advise that the manuscript be proofread once more to remove the present grammatical and spelling errors.
Ex. Line 123: "extent" instead of extent, Line 147: "important for the functioning of the TSP", etc.